# Determinants and drivers of young children's diets in Latin America and the Caribbean: Findings from a regional analysis

**Franziska Gassmann**[1], **Richard de Groot**[2]*, **Stephan Dietrich**[1], **Eszter Timar**[1], **Florencia Jaccoud**[1], **Lorena Giuberti**[1], **Giulio Bordon**[1], **Yvette Fautsch-Macías**[3], **Paula Veliz**[3], **Aashima Garg**[4], **Maaike Arts**[3]

1 UNU-MERIT, Maastricht University, Maastricht, Netherlands, 2 Independent Consultant, Oosterhout, Netherlands, 3 UNICEF Latin America and Caribbean Regional Office, New York, New York, United States of America, 4 UNICEF, New York, New York, United States of America

* ral.degroot@gmail.com

**Data Availability Statement:** Data used in the manuscript to construct Figs 2–4 are publicly available through data.unicef.org. For sake of

## Abstract

The Latin America and Caribbean region exhibit some of the lowest undernutrition rates globally. Yet, disparities exist between and within countries and countries in the region increasingly face other pressing nutritional concerns, including overweight, micronutrient deficiencies and inadequate child feeding practices. This paper reports findings from a regional analysis to identify the determinants and drivers of children's diets, with a focus on the complementary feeding window between the age of 6–23 months. The analysis consists of a narrative review and descriptive data analysis, complemented with qualitative interviews with key informants in four countries: Guatemala, Paraguay, Peru and Uruguay. Findings indicate that poverty and inequality (disparities within countries by wealth and residence), unequal access to services, inadequate coverage of social programmes and lack of awareness on appropriate feeding practices are important drivers for inadequate diets. We conclude that countries in the region need to invest in policies to tackle overweight and micronutrient deficiencies in young children, considering inequalities between and within countries, enhance coverage of social protection programmes, improve coordination between sectors to improve children's diets and expand the coverage and intensity of awareness campaigns on feeding practices, using iterative programme designs.

## 1. Introduction

Nutritious, safe and diverse food is one of the most essential building blocks of human life. Hunger and malnutrition not only violate the basic human right to food but can also have severe consequences that last over one's lifetime. Malnutrition in young children has detrimental and long-lasting consequences for their physical and cognitive development [1]. Throughout the last decades, countries have made remarkable progress towards eradicating child malnutrition, but many children around the world are still at risk of not meeting their dietary needs. While the global share of children under five with stunting (being too short for

transparency, we have included the data needed to generate Figs 2–4 as well as the figures in S2 Text as a supplementary file to the manuscript. Primary data collection for this study involved in-depth interviews with selected key informants. Although ethical approval was not required for this type of study, as employees or affiliates of Maastricht University, the authors were bound by the Netherlands Code of Conduct for Research Integrity (https://www.nwo.nl/en/netherlands-code-conduct-research-integrity). This Code encourages researchers to make their research findings and data public upon completion of research, while establishing valid reasons should this not be possible. The Code lists 'confidentiality' as a valid reason for non-disclosure of research data. In our consent procedure (now included in our submission as appendix C), we assured informants that their responses will not be traceable to them personally by stating: "Your name and other information that could identify you will not be used on any study documents. In the reports and other publications that we make about this study, we may use a quote from your interview but we will not identify who you are." Hence, we are not able to make these transcripts public due to ethical reasons. In addition, we are not able to make (excerpts) of transcripts available to interested outside parties on request since our consent procedure also assured participants that "We will not share your answers with anyone beyond the research team." That being said, we have included selected excerpts of the interviews (quotes) in the current manuscript, section 3.2, and excerpts will be included in the full study report to be published by UNICEF in the near future.

**Funding:** This work was funded by a grant from the the Government of The Netherlands (grant number SC189903). The funder did not play any role in the study. The funds were provided to UNICEF New York HQ, who then distributed the funds across all UNICEF regions based on proposals and needs.

**Competing interests:** The authors have declared that no competing interests exist.

one's age) has declined steadily since 1990, one in five children still experienced impaired growth due to poor nutrition in 2020 [2].

Although the Latin American and Caribbean (LAC) region registers one of the lowest rates of stunting and wasting (being too thin for one's height) globally, regional aggregates hide large disparities across countries [3]. In Guatemala, 47% of children under five are stunted, and in Ecuador, Haiti and Honduras more than 20% of the children suffer from wasting [3]. Meanwhile, adequate maternal, infant and young child nutrition remains an exception rather than the norm. According to UNICEF's latest estimates, a little over one out of three (38%) infants under 6 months old are exclusively breastfed [3]. Although the vast majority (84%) of infants are introduced to solid, semi-solid or soft foods at the recommended age between 6–8 months, many children are consuming solid foods before the recommended 6 months [4]. There is also growing evidence that a dietary transition is taking place and the consumption of foods high in sugar, salt and fat is increasing [5,6], resulting in increasing rates of overweight and its severe form obesity [7]. As a result, 7.5% of children under 5 in the LAC region are affected by overweight, ranging from 3.7% in Haiti to 12.9% in Argentina [2]. Given the large share of children who suffer from micronutrient deficiencies (e.g. iron and vitamin A) [3], LAC faces a 'triple burden of malnutrition' (the co-existence of undernutrition, micronutrient deficiencies and overweight).

Against this background, it is imperative to understand determinants and drivers of children's diets and to identify effective interventions that improve the diets of young children. This is particularly relevant since progress against the key nutritional Sustainable Development Goals (Target 2.1 on ending hunger and Target 2.2 on eliminating malnutrition) has been slow [8]. The main aim of this paper is therefore to analyze the determinants and drivers of young children's diets in LAC. In doing so, we start by providing an overview of current rates of malnutrition and complementary feeding practices in the region. In addition, we aim to identify existing strategic actions that improve young children's diets in the region. Using multiple research methods, including a narrative review, descriptive data analysis and interviews with key stakeholders, the analysis in this paper identifies barriers and opportunities on how to improve the diets of young children in LAC. The focus is on children from 6 to 23 months of age, which is the age when children transition from exclusive breastfeeding to age-appropriate complementary feeding. This period is a critical window in a child's development and if complementary foods and feeding practices are inappropriate, there is increased risk of undernutrition (stunting, wasting), micronutrient deficiencies and overweight [9]. For instance, complementary foods high in sugar and fats can lead to overweight while also failing to meet a child's micronutrient needs. This window is not only critical in meeting children's dietary needs for their growth and development, but it is also when children's food preferences and dietary habits are shaped for the rest of their lives. This is the time when they learn to listen and respond to cues of hunger and satiety, which are essential for upholding healthy diets and weight throughout life.

The analysis is based on the UNICEF action framework to improve the diets of young children during the complementary feeding period [10] (Fig 1). The framework recognizes the role of a situational analysis of the determinants of children's diets. The determinants of young children's diets during the complementary feeding period include adequate complementary foods, adequate complementary feeding practices, and adequate services. Access to adequate food includes physical and economic access to nutritious, safe and affordable foods. These determinants are shaped by context-specific factors–referred to as drivers. Together, they determine children's ability to enjoy nutritious, safe, affordable and sustainable diets that protect, promote and support survival, growth and development.

**Fig 1. The UNICEF action framework to improve the diets of young children during the complementary feeding period.** Source: UNICEF [11].

The action framework reinforces the need to deliver context-specific strategic actions through multiple systems that have the potential to deliver nutrition interventions: the food system, the health system, the water and sanitation system and the social protection system. The food system includes all actors and steps to get food from farm to mouth. It therefore has a critical role in the availability, accessibility and affordability of nutritious food [3]. The health system too plays a key role in child nutrition from pregnancy through the first years of life through access to affordable and high-quality preventive and curative health care. For example, contact with proper health services through antenatal care monitoring can assist women in keeping proper nutrition during their pregnancy. Having received antenatal care (ANC) services in low- and middle-income countries is associated with improved birth outcomes and longer-term reductions of child mortality and malnourishment [12]. Furthermore, delivery in a health facility with a skilled provider—particularly care delivered in public sector facilities—appears to be positively correlated with favorable breastfeeding practices, providing a good foundation for nutrition at birth [13]. After birth, growth monitoring and nutrition counselling through the health system can be effective ways to improve nutritional status as well as providing an entry point to preventive and curative health care [14]. Water, sanitation and hygiene (WASH) systems are particularly important for safe food preparation, drinking water, and the reduction of infectious disease (like diarrhea) that would limit children's absorption of micronutrients. The lack of access to proper WASH services can affect child nutrition through diarrheal diseases, intestinal parasite infections and environmental enteropathy [15]. Social protection systems can support households in meeting their dietary needs in different ways. Direct assistance in the form of cash transfers can help caregivers to buy healthy complementary foods for their children, thereby playing an essential role in facilitating access to appropriate diets [16]. Alternatively, transfers can help in seeking health care or upgrading housing conditions, affecting other determinants of malnutrition [17].

## 2. Methods

### 2.1. Ethics statement

This study is primarily based on secondary data analysis and a desk review of relevant documents and journal articles. Primary data collection for this study involved in-depth interviews with selected key informants. Participation in the interview was voluntary and consensual. Prior to the interview, written consent was sought and this was reaffirmed at the beginning of

the interview. Although ethical approval was not required for this type of study, as employees or affiliates of Maastricht University, the authors were bound by the Netherlands Code of Conduct for Research Integrity [18].

## 2.2. Methods

This regional analysis is based on a narrative review and a descriptive analysis of secondary nutritional data, complemented by qualitative interviews with key informants in four selected countries (Guatemala, Paraguay, Peru and Uruguay).

We conducted a descriptive analysis of malnutrition rates (stunting, wasting and overweight) and key complementary feeding indicators for assessing infant and young child feeding practices following WHO-UNICEF infant and young child feeding (IYCF) indicators guidelines using publicly available data, as shown in Table 1. The data for malnutrition rates was extracted from the UNICEF/WHO/World Bank Joint Malnutrition Estimates Expanded Databases, which is the prominent source for malnutrition data to monitor progress towards the Sustainable Development Goal, Target 2.2 to end all forms of malnutrition [2]. Infant and young child feeding indicators were extracted from the UNICEF Global Database on Infant and Young Child feeding, the key source for nutritional data on children [19]. Data visualizations were prepared with Microsoft Excel.

The narrative review was organized around two elements of the UNICEF Action Framework: 1) The situation and status of young children's diets, including what, when and how they are fed and the determinants and drivers of young children's diets, especially those related to food, services and practices; and 2) Strategic actions to improve young children's diets delivered through the food, WASH, health and social protection systems. Documents were identified via several approaches.

The objective of the narrative review was to 1) identify patterns and determinants of young children's diets in the region, 2) understand the economic and socio-cultural contexts and drivers relevant to child feeding in LAC, 3) map out policy and existing approaches to improving young children's diets, and 4) identify knowledge gaps and directions for future research and policy action in the region. The review involved three stages and was conducted in January and February 2020.

First, several key research reports and grey literature were reviewed. This included scoping and review reports on the topic from international organizations such as UNICEF, the FAO,

**Table 1. Overview of key infant and young child feeding indicators and their definitions.**

| Indicator | Definition |
|---|---|
| Exclusive breastfeeding | The % of children aged 0–5 months who are exclusively breastfed. |
| Introduction of solid foods | The % of children aged 6–8 months who receive solid, semi-solid or soft foods. |
| Minimum dietary diversity (MDD) | The % of children aged 6–23 months who received foods from at least five food groups from the following eight: 1. Breastmilk, 2. Grains, roots, tubers, 3. Legumes and nuts, 4. Dairy products (formula, milk, yogurt, cheese), 5. Flesh foods (meat, fish, poultry, liver, organ meats), 6. Eggs, 7. Vitamin-A rich fruits and vegetables, 8. Other fruits and vegetables |
| Minimum meal frequency (MMF) | The % of children aged 6–23 months who receive solid (solid, semi-solid or soft) foods the minimum number of times. The minimum number of times of complementary feeding for breastfed children is: 2 times for those aged 6–8 months, 3 times for those aged 9–23 months. The minimum number of feeding for non-breastfed children is 4 times solid foods and/or milk feeds for 6–23 months of age. |
| Minimum acceptable diet (MAD) | The % of children aged 6–23 months who receive both the minimum dietary diversity and the minimum meal frequency. |

Source: World Health Organization & UNICEF [20].

and WFP. The purpose of these reports was to provide a preliminary understanding of recommendations, indicators, and best practices related to the complementary feeding of young children. Key words to be used as search strings in the second stage of the scoping review were also identified based on these reports.

The objective of the second stage of the review was to identify academic and grey literature on the state, determinants and drivers of young children's diets in the region. This stage relied on a protocol for searching and including literature. The team searched for academic literature via four academic databases/search engines: Google Scholar, PubMed, LATINDEX and Elsevier by using search terms (including BOOLEAN terms) such as "complementary feeding", "complementary food", "child diet", "child nutrition", "infant nutrition", "exclusive breastfeeding", "responsive feeding", "food system" etc. in combination with "Latin America", "Caribbean", and names of countries in the region. The search terms were entered in English, Spanish and Portuguese. We also searched on websites of relevant organizations and initiatives (e.g. Food and Agricultural Organization of the United Nations, the World Food Programme, UNICEF, the International Food Policy Research Institute, the Inter-American Development Bank, the Institute of Nutrition of Central America and Panama, the Global Alliance for Improved Nutrition, Scaling up Nutrition etc.). Finally, we used snowballing techniques starting from key documents identified through the desk review, in which the reference list in these key papers was traced and relevant papers reviewed. These reference lists were subjected to the same key search terms as above to identify potentially relevant papers. To provide the most up-to-date information, the literature search was limited to documents published after 2010. The search protocol did not include restrictions on methodology or sample size: both quantitative and qualitative research, as well as review papers were permitted if they aligned with the study objective. A geographic restriction was applied: publications had to pertain to the region (either as a whole or to individual countries in the region). See S1 Text for details on the search protocol.

The literature identified through these stages were first subjected to title- and abstract-screening. The information of those that appeared relevant were entered into an Excel sheet, and the study team conducted a full-text screening to decide which publications to include (based on their relevance and quality).

To gain a more in-depth understanding of the determinants and drivers of children's diets as well as institutional contexts, key informant interviews were conducted with stakeholders in four countries: Guatemala, Paraguay, Peru and Uruguay. Countries were selected purposefully to include a broad range of dietary and nutrition situations in the region, as well as different socio-economic and policy situations. For example, Guatemala has the highest stunting rates in the region and a high cultural diversity. Paraguay also has a diverse population. Peru is considered a success story in reducing stunting. And while Uruguay is a high-income country; the others are all upper-middle income according to the World Bank classification [21]. In each of the four countries, four or five key informants from different sectors and institutional levels were selected, including staff from NGOs, government officials, researchers and international development practitioners. Recruitment into the study was based on purposeful selection in consultation with local UNICEF offices. By purposefully selecting informants based on their expertise, the study was able to generate the most salient information for each country within a relatively short timeframe.

Interviews followed a semi-structured approach and respondents were asked about the overall situation of child nutrition in their countries, about the barriers and facilitators of improving children's diets, and the legal and policy frameworks. Interview questions were adjusted for the specific type of respondent. For example, interviews with central government bodies were more concerned with the overall situation and the government's priorities and

vision. NGO workers were asked about their concrete experiences on the ground, descriptions of children's diets and needs, as well as potential barriers and lessons learned. Interviews were conducted in July and August 2020 via telephone or over the internet, in the language preferred by the respondent (Spanish or English). Interviews were completed by three members of the research team, while the analysis was conducted by two other researchers to avoid bias in the interpretation of the interviews. All interviewers had no relation to the key informants prior to the interview. The interviews adhered to ethical standards as set out by the Netherlands Code of Conduct for Research Integrity [18]. Participation was voluntary and consensual. Prior to the interview consent was sought and this was reaffirmed at the beginning of the interview. In addition, provided the participant consented, the interviews were audio-recorded to ensure the statements were captured in the correct context. These audio recordings were stored in a password protected online server and were only accessible to the interviewers to clarify any statements during the analysis stage. Furthermore, the data was treated confidentially and anonymously, and hence no direct reference to the key informant is made in this paper. The interviewers, who were also fluent in English, took detailed notes during the interview and then translated the notes into English for the analysis. Additional information regarding the ethical, cultural, and scientific considerations specific to inclusivity in global research is included in the Supporting Information (S4 Text).

The qualitative analysis followed a deductive content analysis process [22], focusing on extracting predetermined key emerging themes across six broad topics: 1) General context in the country regarding the food environment and recent developments, 2) Institutional context and legal frameworks with respect to complementary feeding; 3) Food and agriculture, 4) Health system; 5) Water, Sanitation and Hygiene (WASH); 6) Social protection. The latter four sectors were included following the UNICEF conceptual framework as sectors that have the potential to deliver nutrition interventions at scale. The interviews were analyzed per country and answers were categorized under each relevant topic using a predefined analysis sheet. For each topical area, it was determined to what extent the various informants for each country agreed on the major themes, as a triangulation measure. Based on this categorization, a summary was developed per country, which formed the basis for a more extensive reporting and comparisons on the emerging findings across the four countries. We integrated findings from the narrative review and the descriptive analysis into the country qualitative analysis to triangulate findings from the expert interviews with the available data and country-specific literature.

## 3. Results

On average, the LAC region registers one of the lowest rates of stunting (11%) and wasting (1.3%) globally [2]. However, notable exceptions remain hidden by regional aggregates: in Guatemala, 47% of children under five are affected by stunting, and Ecuador, Haiti and Honduras have stunting rates above 20%. Meanwhile, overweight affects more and more of the region's population, including children. The magnitude of the problem of overweight is shown in Fig 2. In most countries with available data (collected after 2010), rates of overweight among children under 5 are 5% or higher, with the largest burden in Argentina, Paraguay and Barbados. Malnutrition in children under 5 also differs within countries, with stunting and wasting more prevalent among children in the lowest wealth quintiles and (for stunting) in rural areas (S2 Text, Fig A, B, D and E). In contrast, overweight is more concentrated among children in urban areas and in the highest wealth quintile (S2 Text, Fig C and F). Recently, micronutrient deficiencies have become a more pressing public health concern than stunting. The rate of micronutrient deficiency, in terms of vitamin A and iron deficiency, in children

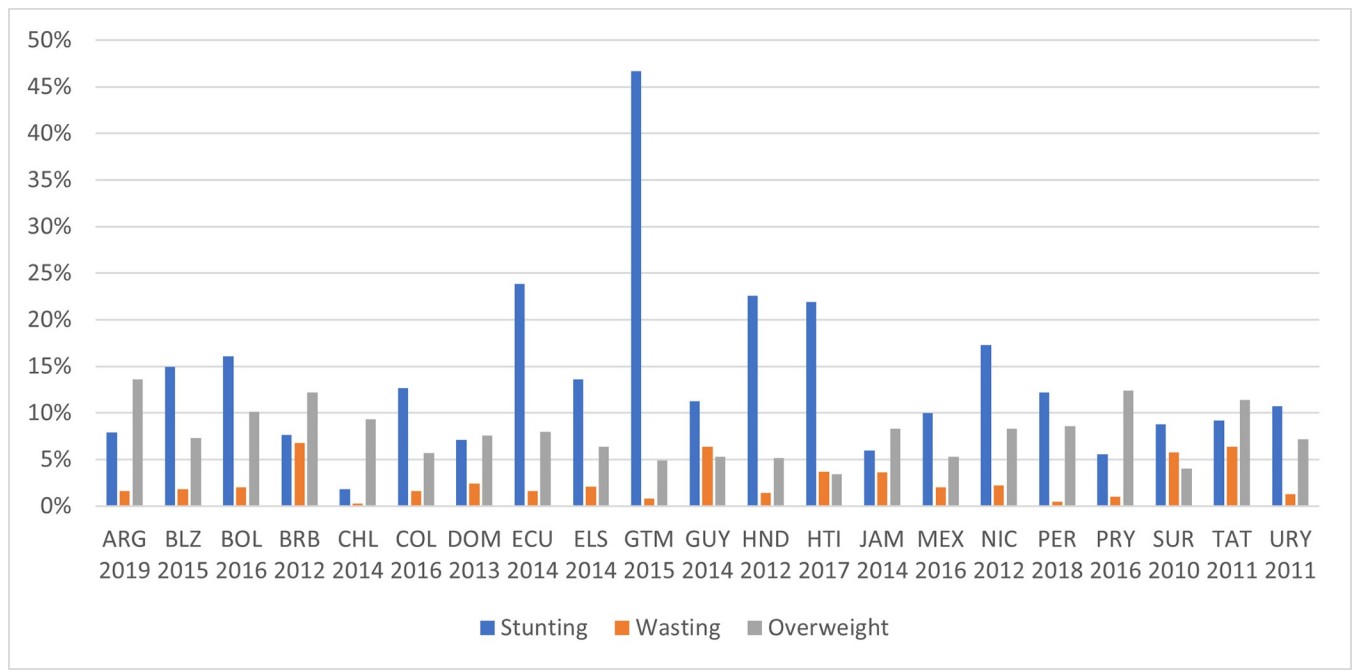

**Fig 2. The prevalence of stunting, wasting and overweight in Latin American and the Caribbean countries.** Note: The figure shows the prevalence of stunting, wasting and overweight for each country using the latest available data since 2010. Abbreviations: ARG: Argentina, BLZ: Belize, BOL: Bolivia, BRB: Barbados, CHL: Chili, COL: Colombia, DOM: Dominican Republic, ECU: Ecuador, ELS: El Salvador, GTM: Guatemala, GUY: Guyana, HND: Honduras, HTI: Haiti, JAM: Jamaica, MEX: Mexico, NIC: Nicaragua, PER: Peru, PRY: Paraguay, SUR: Suriname, TAT: Trinidad and Tobago, URY: Uruguay. Source: authors' calculations based on UNICEF Global Database.

under 5 is 36%, 37% and 46% in South America, Central America and the Caribbean, respectively [3].

An overview of the key complementary feeding indicators paints the picture of a very heterogenous region (Fig 3). Countries with the highest rates of exclusive breastfeeding during the first six months of life are Peru (66%), Bolivia (58%), Guatemala (53%), El Salvador (47%) and Haiti (40%). Among countries with available data, Suriname (3%) and the Dominican Republic (5%) lag the most in terms of exclusive breastfeeding. Within countries, exclusive breastfeeding is more common among households in the lowest wealth quintile and those in rural areas (S2 Text, Fig G and I).

Most infants between six and eight months of age received solid, semi-solid or soft foods in all countries with available data (Fig 3). In Argentina, Cuba, El Salvador, Haiti and Peru, over 90% of infants have been introduced to complementary foods during this period. While on average, most infants are receiving solid, semi-solid or soft foods at the recommended time, there is quite some regional variation within the region. For example, in Belize, Ecuador, Guatemala, Jamaica, Panama, Suriname and Trinidad and Tobago, the corresponding rate was below 80%. There is also variation within countries, with a larger share of urban children and those in the highest wealth quintile receiving solid, semi-solid or soft foods at the recommended time (S2 Text, Fig H and J).

Information about the indicators on dietary adequacy (MDD, MMF and MAD) is limited in the region (Fig 4). Some countries have data on one indicator but not the other, which makes it difficult to measure compliance with minimum acceptable diet recommendations. In the countries with available data, the rates of young children meeting the MDD requirements are alarming. Most countries reported rates below 60%, with the most alarming situation in

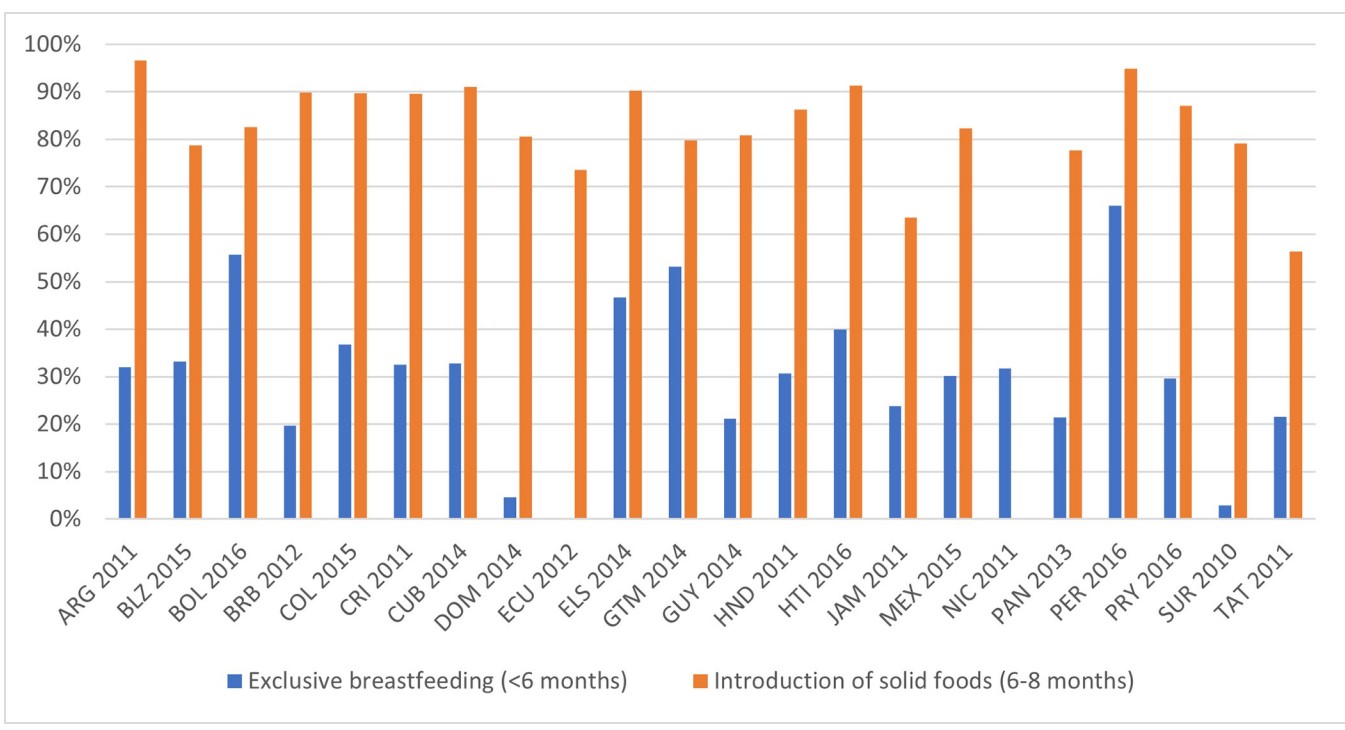

**Fig 3. Exclusive breastfeeding and introduction of solid foods in Latin American and the Caribbean countries.** Note: The figure shows the prevalence of exclusive breastfeeding among children < 6 months and the share of infants aged 6–8 months who are introduced to solid foods for each country using the latest available data since 2010. Abbreviations: ARG: Argentina, BLZ: Belize, BOL: Bolivia, BRB: Barbados, COL: Colombia, CRI: Costa Rica, CUB: Cuba, DOM: Dominican Republic, ECU: Ecuador, ELS: El Salvador, GTM: Guatemala, GUY: Guyana, HND: Honduras, HTI: Haiti, JAM: Jamaica, MEX: Mexico, NIC: Nicaragua, PAN: Panama, PER: Peru, PRY: Paraguay, SUR: Suriname, TAT: Trinidad and Tobago. Source: authors' calculations based on UNICEF Global Database.

Haiti, where only one out of five infants (19%) met the requirement for a diverse diet. Peru (82.9%) has the highest share of children receiving a sufficiently diverse diet. For those countries for which multiple years of data are available, trends are not positive. In the Dominican Republic, the rate of MDD decreased between 2007 and 2014 from 58% to 51%. In Haiti, the rate decreased from 23% in 2005 to 19% in 2016, and in Guyana, the rate decreased from 48% in 2009 to 40% in 2014. In Honduras, the rate of MDD increased only marginally from 58% to 61% between 2005 and 2011. The exception is Peru, which registered an increase from 73% in 2007 to 83% in 2016, though the trend stagnated over the last three data periods.

Compliance with MMF recommendations is less of a challenge in the region. Two Caribbean countries, Haiti (39%) and Jamaica (42%) report the lowest prevalence of MMF. In all other countries, at least half of young children are fed the minimally recommended number of times, reaching at least 80% in the Central American countries of El Salvador, Guatemala, Honduras and Mexico.

The computation of MAD requires data on both MDD and MMF, it can therefore only be reported for 10 countries (Fig 4). There are large differences in the prevalence of MAD. In El Salvador nearly two-thirds of children 6–23 months receive a minimal acceptable diet. In Cuba, Guatemala and Honduras, every second infant receives both enough quantity and types of food, while in Haiti only one in nine does.

Indicators of dietary adequacy vary considerably with wealth and residence. A larger share of children in wealthier households and those residing in urban areas meet dietary recommendations compared to children in the lowest wealth quintile or those in rural areas (S2 Text, Fig K-P).

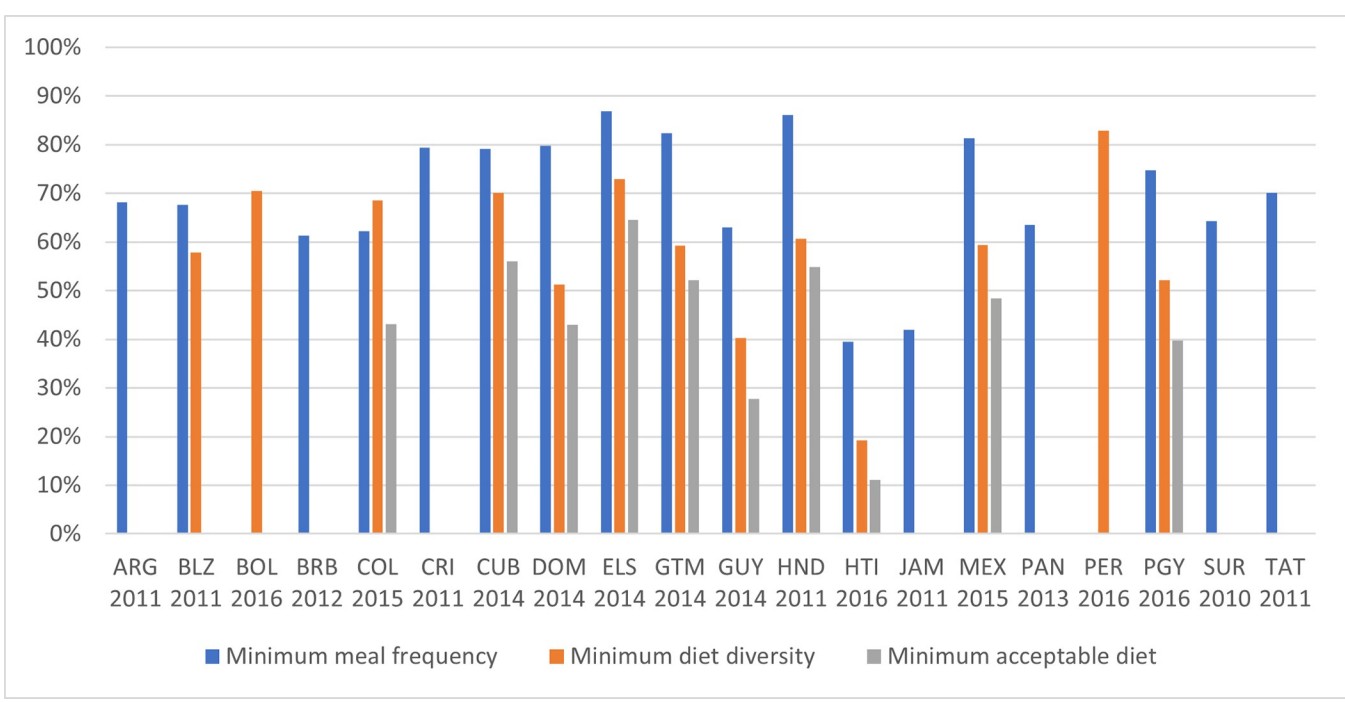

**Fig 4. Adequacy of children's diets in Latin American and the Caribbean countries.** Note: The figure shows the prevalence of minimum meal frequency, minimum diet diversity and minimum acceptable diet among children 6–23 months for each country using the latest available data since 2010. Abbreviations: ARG: Argentina, BLZ: Belize, BOL: Bolivia, BRB: Barbados, COL: Colombia, CRI: Costa Rica, CUB: Cuba, DOM: Dominican Republic, ELS: El Salvador, GTM: Guatemala, GUY: Guyana, HND: Honduras, HTI: Haiti, JAM: Jamaica, MEX: Mexico, PAN: Panama, PER: Peru, PRY: Paraguay, SUR: Suriname, TAT: Trinidad and Tobago. Source: Authors' calculations based on UNICEF Global Database.

### 3.1. Determinants and drivers of children's diets in Latin America and the Caribbean

**3.1.1. Access to adequate foods.** The LAC region produces more than enough quantity of food to meet the energy needs of its population [23]. Home production in the form of family farming plays a major role, accounting for about 81% of agricultural activities in LAC [24], and supplies between 27% and 67% of food production [5]. The growth of food production exceeds population growth and the region is blessed with favorable environmental conditions and capacity for production. Marked differences exist between and within sub-regions, which makes trade and collaboration between countries important for ensuring equitable food availability [23].

At the household-level, a major obstacle to consuming diverse and healthy diets is the inability to afford them. The cost of a simple, healthy plate of food (a stew made of beans or other pulses, paired with a carbohydrate component that matches local preferences amounting to 600 kcal) is estimated to cost 4.0%, 5.4%, 6.3% and 35% of an average daily income in Guatemala, El Salvador, Nicaragua and Haiti, respectively. A similar plate of food would cost only 0.6% of the average daily income of a citizen in New York. Since these estimates are averages, costs are likely higher for poorer households [25]. Another study on the relative costs of nutritious food showed that dark green leafy vegetables and vitamin-A rich fruits and vegetables are a relatively expensive source of calories in LAC. On the other hand, with the exception of fish, animal-sourced foods like diary and meat are relatively inexpensive [26]. A cost-of-diet analysis in El Salvador showed that between 24% and 30% of households would not be able to purchase a minimum cost nutritious diet [27].

At the child-level, grains, roots and tubers are the most prevalent complementary foods in the region, consumed by 88% of infants [4]. Animal source foods are also relatively widespread, with dairy products being the most common, followed by flesh foods and eggs. Overall, less than 10% of infants were reported to not consume any type of animal source food [4].

There are considerable disparities in the diets of infants across areas of residence and quintiles of wealth, with animal source food consumption being more common in urban and rich households [4]. An understanding of this inequality is important because animal source foods are the richest sources of iron and zinc [28], and meat intake is associated with better growth and development outcomes [29].

As countries move forward on the path of socio-economic development, formula-feeding becomes more prevalent [3]. Recent aggregate sales data confirms that commercial milk-based formulas are increasingly becoming a part of young children's diets in the region. Between 2008 and 2013, Brazil and Peru were among the countries with the fastest growing sales of formulas globally by 133% and 160%, respectively. In contrast, the sales of infant/child milk-based formulas grew by only 11% in Mexico [30].

Overweight, obesity and the shift towards unhealthy diets can also be explained by the increased availability of energy-rich but micronutrient poor, processed foods in the region. Recently, there has been a major shift in the availability and consumption of ready-to-eat, ready-to-heat, processed, and packaged foods and beverages in the LAC region [23,31]. For example, sales of ultra-processed food and drink products increased by 8.3% between 2009 and 2014, with these trends projected to continue to grow [23]. In addition, the consumption of ultra-processed products grew by more than 25% between 2000 and 2013, and fast-food consumption increased by almost 40% over the same period. In Latin America, most ultra-processed products are increasingly sold in convenience stores, supermarkets, and hypermarkets [6].

**3.1.2. Adequate services.** Some countries in the region still have a long way to go to achieve adequate WASH services for all, especially Bolivia and Haiti. While in 2015, 95% of the region's population used an improved drinking water source, the gap between the poor and the better-off, and between rural and urban areas, remains wide [32,33].

The significance of inappropriate WASH services for children's diets was highlighted by a qualitative study of 16 low- and middle-income countries (including Bolivia, Colombia, Guatemala, Haiti and Mexico), which examined the relationship between water security and infant feeding [34]. Water insecurity was associated with poor quality and quantity of drinking water available for young children. Moreover, caregivers echoed that unsafe water meant that they could not prepare complementary foods as safely and hygienically as they would have wished to. Some of them substituted preferred complementary foods with other, less preferred ones to reduce the risk of contamination. Some respondents also reported delaying infant feeding until clean water is available. Respondents also observed a link between poor water quality and infant morbidity (such as infectious diseases, diarrhea, stomach and skin irritation, undernutrition and dehydration).

Overall, 34% of all households in the region are covered by social assistance programmes, including 60% of all households in the bottom quintile of the income distribution [35]. The LAC region has been a global leader in demonstrating the potential of cash transfer programmes in supporting the health and nutritional needs of young children, with different studies identifying 15 to 30 national conditional cash transfers (CCTs) in 2015/16 [36,37]. In 2016, 16.9% of all households (a total of nearly 30 million families) in the LAC region participated in at least one programme [36]. A review of three CCTs in Brazil, Colombia and Mexico found consistent evidence that these programmes had a positive effect on child nutrition [38].

In terms of health services, a regional study showed that out of 15 countries in LAC, 13 have growth monitoring policies in place [39], but it is unclear at a regional level how many

children comply with monitoring visits. Moreover, Community Health Workers (CHWs) can play a key role in nutrition service delivery through the health system. For example, a case study in Haiti showed that CHWs performed 24 of the recommended 38 nutrition services [40]. Yet, little information is available across the region on the numbers of CHWs, and their responsibilities [41].

**3.1.3. Adequate feeding practices.**   Indigenous populations are often more at risk of malnutrition and poor health outcomes [42,43]. A study of infant feeding practices in the Peruvian Amazon found that traditional complementary foods in the Ajawún community largely met WHO recommendations, and the foods consumed were more nutrient dense and higher quality than marketed foods. However, the intake of some micronutrients (mostly zinc, iron and Vitamin A) was below the adequate level [44]. Another study, also conducted in the Peruvian Amazon, concluded that the period of exclusive breastfeeding in the community was too short and that meal frequency and diversity did not meet standards [45].

Caregivers' perceptions and preferences for complementary foods may diverge from those of nutritionists' and policymakers [46]. Researchers in Mexico found that mothers working outside the home value the convenience, flavor and vitamin content of marketed and/or processed complementary foods but are also concerned about their sugar and chemical contents. The shift towards formula feeding is also related to their convenience, especially in the absence of daycare, maternity leave and flexible work policies [30].

**3.1.4. Existing strategic actions to improve children's diets.**   In LAC, the supplementation of food with vital micronutrients, such as vitamin A, iron, zinc and others is widespread [47]. Many countries also provide universal fortification of staple foods, for example iodine in salt, vitamin A in sugar, and varying fortifications including iron, zinc, vitamin B12, folic acid and more in wheat and maize flour and rice [47]. Twelve countries target households with infants 6–23 months with multiple micronutrients powders so that complementary foods can be fortified by caregivers themselves [47].

Several countries have adopted regulatory strategies to reduce the accessibility, availability and desirability of sugar-sweetened beverages and unhealthy or ultra-processed foods. These strategies include 'sugar taxes', advertising restrictions and requirements for transparency in food labeling and facilitating purchasing choices by caregivers. An overview of Latin American countries found 39 such regulatory strategies aimed at reducing the consumption of unhealthy foods and beverages, with Chile, Ecuador and Mexico having made the most comprehensive efforts [48].

Nutrition actions are most commonly delivered through food security and nutrition policies, but some countries have incorporated nutrition goals in other sectors such as agriculture, education, environment, development and employment policies [39]. A review of nutrition policies showed that most of the 18 countries reviewed have one or more nutrition-related and/or sectoral policy in line with the WHO recommendations for Comprehensive Implementation Plan on Maternal, Infant and Young Child Nutrition [39]. Interdisciplinary evidence on what types of national strategies are the most effective, however, are lacking [48].

## 3.2. A view from the experts

Interviews with key informants provided further insights into the determinants and drivers of infant and young child nutrition in four countries: Guatemala, Paraguay, Peru and Uruguay. The affiliation of key informants per country is provided in Table 2. Names and positions of the informants are excluded to protect their identity. Key findings from the interviews are summarized in Table 3 and described below.

Child undernutrition remains a widespread issue in Guatemala: the country has the highest rate of stunting in the LAC region, affecting nearly half of all children under five, and almost

**Table 2. Affiliation of key informants in selected countries.**

| Country | Organization |
| --- | --- |
| Guatemala | Secretaría de Seguridad Alimentaria y Nutricional |
| | FAO |
| | UNICEF |
| | Helvetas |
| Paraguay | Instituto Nacional de Alimentación del Ministerio de Salud |
| | FAO |
| | Sociedad Científica y Academica |
| | Sociedad Científica y Academica |
| Peru | Ministry of Development and Social Inclusion |
| | CENAN |
| | WFP |
| | FAO |
| | Instituto de Investigación Nutricional |
| Uruguay | Uruguay Crece Contigo |
| | formerly UNICEF |
| | Uruguay National University |
| | Uruguay National University, FAO |

half of all children under six suffer from anemia. There appears to be inequality of access to healthy diets between different groups of the population, evidenced by the concentration of nutritional deficiencies among children in low-income households, in rural areas and from indigenous groups. In terms of adequate food, low dietary diversity is one of the main drivers

**Table 3. Summary of drivers of children's diets per determinant based on key informant interviews, by country.**

| Determinant | Guatemala | Paraguay | Peru | Uruguay |
| --- | --- | --- | --- | --- |
| **Adequate food** | • Lack of resources and limited caregiver awareness results in poor diet diversity<br>• Lack of access to iron-rich food | • Low economic access to food<br>• High availability of ultra-processed foods<br>• High relative cost of fruits and vegetables | • High cost of animal-source foods<br>• Strong promotion of breastmilk substitutes<br>• In response to concerns over anemia, the government and commercial parties have promoted the use of iron-rich products for infants, compromising fresh and healthy alternatives | • Purchasing power of the household<br>• Availability and convenience of ultra-processed foods |
| **Adequate services** | • Unequal access to health services, due to geographical barriers<br>• Machismo and social restrictions are substantial barriers to access to health care<br>• Funding and managing access to WASH<br>• Lack of nutrition-sensitivity in social programmes | • Barriers for access to healthcare services include distance, and lack of trained personnel in primary care facilities.<br>• Limited coverage of social protection programmes, that have limited coverage due to budgetary restrictions<br>• Inequalities in access to WASH services.<br>• Limited support regarding infant and young child feeding | • Lack of trained health staff<br>• Inadequate flow of budgetary resources and lack of coordination to support the regional directorates of Health especially in rural areas<br>• Ineffective, vertical relationship between health personnel and caregivers<br>• Lack of urban social protection coverage<br>• Poor access to potable water and lack of refrigeration, particularly in rural areas | • Outdated health professionals' knowledge on infant and young child feeding<br>• Mistrust towards health professionals and their recommendations<br>• Limited coverage of social protection programmes, with demand exceeding supply |
| **Adequate practices** | • Cultural beliefs in the use of chlorine<br>• Traditional gender roles around hygiene<br>• Lack of knowledge and nutrition-related awareness for mothers | • The food culture results in a diet that is high in fat and carbohydrates<br>• Early introduction to complementary foods<br>• Early introduction to sweet beverages | • Lack of knowledge among caregivers<br>• Convenience to use processed foods among working urban households | • High consumption of highly processed foods, especially in children<br>• Beliefs and social representations around food<br>• Limited time for food preparation |

of nutritional deficiencies, according to key informants, driven by a lack of resources, limited caregiver awareness and lack of access to iron-rich foods

*"70 percent of people and workers in the agricultural sector have children that are suffering from stunting. They don't have enough resources. This is one of the underlying problems."* (Staff from an international organization (IO))

Adequate services are limited by unequal access to quality health care due to geographical barriers and traditional gender roles. In addition, funding and management of WASH services is hampered by structural issues such as financing and administrating the technical services. Moreover, social protection programmes currently lack nutrition sensitive goals.

Adequate practices in Guatemala are constrained by cultural beliefs, for example on the use of chlorine to treat water. In addition, traditional gender roles around hygiene persist and men do not follow standard hygiene practices.

Despite a strong policy framework, in the form of the recently established *Gran Cruzada Nacional por la Nutrición*, there is room for improved inter-institutional collaboration and continuity over time in governments' actions, according to the key informants.

*"This [nutrition policies] changes a lot, depending on the government in power. So I think there has to be continuity. Simultaneously, the diversity of the diet, raising animals, having family agriculture, having a complementary feeding program, continuing with powdered micronutrients and the theme of behavior change should be promoted. This has to be an inter-institutional work, where many ministries collaborate towards the same message, that is, with integrated actions of the Government."* (Government official)

In Paraguay, overweight among young children is a more pressing public health problem than undernutrition. Only three out of ten children under six months of age are exclusively breastfed and children's diets are low in diversity. Major barriers to adequate food include poverty, the price of healthy products versus processed foods and the high availability of ultra-processed foods (Table 3).

*"The cost of the food that can provide you with a nutritionally appropriate diet is very high in relation to a cost that provides you with the same calories, but in a totally unbalanced way, with excess carbohydrates or processed foods"* (Staff from IO)

In terms of adequate services, key informants noted barriers to access healthcare services, including geographic distance as well as a lack of trained health staff. Moreover, social protection programmes, like PANI and *Tekopora* only provide limited coverage, due to budgetary restrictions. The country also exhibits inequalities in access to WASH services. While 95% of households in Paraguay have access to an improved source of drinking water, and 83% has access to an improved sanitation facility, there are large inequalities by area of residence, income level and for indigenous groups [49]. There is also limited support in the country regarding infant and young child feeding according to key informants, resulting in poor caregivers' knowledge and practices about how to feed young children.

Furthermore, key informants noted that adequate practices in Paraguay are limited by a food culture that is high in fat and carbohydrates, the (too) early introduction of complementary foods and early introduction of sweet beverages, like juices, at very early ages.

There is also limited attention to the issue of overweight and accompanying policy measures on the political agenda, for example by regulating ultra-processed foods, according to

key informants. Experts also noted the need for strong coordination between different State and non-State actors, as the issue of child nutrition extends across sectors:

*"We must strive to achieve that common vision and join forces to meet the proposed goals. Because sometimes along the way we lose strength if we work kind of independently."* (Government official)

Peru made significant progress to reduce stunting, and it outperforms the other countries in the region when it comes to complementary feeding practices. Yet, overweight and micronutrient deficiencies, notably anemia, continue to be public health problems. Respondents stated the low levels of caregiver's knowledge on appropriate feeding practices, insufficiently trained health workers and ineffective coordination mechanisms between relevant line ministries and other stakeholders as the key bottlenecks for adequate children's diets. Furthermore, animal-sourced foods tend to be expensive, especially in rural areas, and pharmaceutical companies heavily promote the use of breastmilk substitutes. Iron-rich, processed foods are also promoted in response to concerns over anemia, yet this compromises resources for fresh and healthy foods.

In terms of health services, key informants also expressed concerns about the inadequate flow of budgetary resources and the lack of coordination to support regional directorates of health. In addition, according to key informants, health sector personnel have a 'vertical relationship' with clients, giving orders rather than providing tailored recommendations for complementary feeding practices. The social protection landscape is characterized by poor coverage in urban areas, while access to WASH, like potable water and sanitation, is more restricted in rural areas.

Besides lack of nutritional knowledge among caregivers, convenience to purchase processed food was cited as a barrier for adequate practices. In urban areas, many mothers work in informal jobs, creating a barrier for exclusive breastfeeding and adequate complementary feeding. Due to convenience and lack of time to prepare meals, use of processed foods is high among working caregivers in urban areas.

Experts expressed that Peru has a strong legislative and policy framework, in part due to the progress made when the nutrition agenda was under the prime minister's responsibility.

*"The time we had improvements in stunting was the time when nutrition was managed at the prime minister level. Nowadays, nutrition has been delegated to the social inclusion minister, but he does not have the same power to push for results."* (Staff of IO)

A key emerging theme was the need for integrated and cross-sectoral collaboration, as informants recognized that child nutrition is a complex issue reaching across sectors. In the words of one informant:

*"I believe that the greatest obstacle is the ability to articulate between the sectors in the State, because such a reform can only be achieved by the State."* (Government official)

Uruguay's malnutrition rates are in line with the regional average, but the country lacks recent data on complementary feeding practices and micronutrient deficiencies at the time of the study. Key barriers for healthy diets include monetary poverty, the high intake of ultra-processed foods, and the lack of credibility of health workers. Another barrier in the health sector is outdated health professionals' knowledge on infant and young child feeding. According to key informants, specialists often provide conflicting advice, together with a perceived lack of

empathy from their side, which undermines their credibility. As for social protection services, including *Uruguay Crece Contigo*, they are often limited in terms of coverage. According to the key informants, while UCC targets the entire population of pregnant women and children under four years, there is more demand than the programme can meet.

Adequate feeding practices are further limited by a high consumption of ultra-processed foods, especially among children, driven by the strong influence of marketing. In addition, certain beliefs and social representations around food create barriers. For example, the introduction of meat is believed to cause choking, which leads some parents to introduce it at a later age than appropriate. Finally, experts stated that families have a limited amount of time for food preparation and to devote for eating.

Overall, Uruguay's policy landscape to support progress in terms of healthy diets is adequate, according to key informants. However, there is room for the expansion of integrated social programmes and a stronger policy response to unhealthy eating habits.

> *"There is a demand greater than the coverage capacity, with the exception of transfers. In the education centers of public offer, as well as in the family accompaniment in "Uruguay Crece Contigo", we have more demand than supply."* (Government Officers)

Experts expressed that regular data collection and monitoring of complementary feeding indicators should be prioritized to monitor progress. In addition, there needs to be a better vision on how the food system can support health diets.

> *"There is a lack of vision of the food system to guarantee an adequate diet. What are the relationships between the different components of the food systems, in order to guarantee adequate food, not only complementary food, but food and all stages of life?"* (Researcher)

## 4. Discussion

This regional analysis sought to identify determinants and drivers of young children's diets in LAC. It applied the UNICEF Action Framework for improving young children's diets to examine determinants and drivers in terms of adequate food, adequate services and adequate practices. The study also provided a detailed look at nutritional indicators across the region. In addition, the analysis identified strategic actions to improve children's diets through the food system, health system, WASH system and social protection system.

The descriptive analysis showed that the region performs rather well in child undernutrition prevalence, but the dietary transition of the past decades has brought about new challenges. Wasting and stunting have reached low aggregate levels, but sub-regional differences and notable exceptions exist. In addition, disparities within countries exist by residence and wealth status, with those in rural areas and in the poorest quintile in general having poorer nutritional outcomes and complementary feeding practices. Despite progress in stunting reduction, undernutrition (stunting and wasting) and micronutrient deficiencies of young children remain a concern. Moreover, overweight and its severe form obesity are becoming a regional challenge. This triple burden of malnutrition (the co-existence of undernutrition, micronutrient deficiencies and overweight) is perhaps the most pressing issue for countries in LAC to tackle, in order to realize good nutrition and health for all children.

Complementary feeding indicators paint a mixed picture. In countries with low rates of exclusive breastfeeding, too early introduction of complementary foods or beverages is likely to be a concern, particularly if those foods or beverages are high in sugar, sodium or unhealthy fats. Many children are consuming solid foods during the 6-8-month window, but many are

introduced to them too early as demonstrated by varying rates of exclusive breastfeeding in the first six months of life. Too early introduction of complementary foods puts the child at risk of early weaning and compromises the essential nutrients from breastmilk. Indeed, nearly half (48%) of all children aged 4–5 months in LAC received solid foods, suggesting that too early introduction of complementary food is common [50]. Most infants are fed an appropriate number of times a day, but their diets are lacking in diversity. Overall, the concern is the quality rather than quantity of diets.

The low quality of diets is linked to the availability and convenience of packaged, ultra-processed, and sugary foods and drinks. According to key informants, healthy food is often more expensive and more difficult to access than pre-packaged and processed products, especially in urban areas. Access is compounded by a lack of economic means. Research shows that such foods are associated with overweight/obesity and many nutrition-related noncommunicable diseases [6,51]. Evidence from Brazil, Mexico, and Peru suggests that infants' diets are rich in sugar-sweetened beverages and other unhealthy products [44,52,53]. Intense lobbying of the food industry may prevent efficient regulation to prevent high intakes of such ultra-processed foods. The quality of diets also varies by the purchasing power of the household and the area of residence. Infants in poor and rural households are fed less animal source foods than their peers in better-off and urban families. However, there are considerable knowledge gaps about how changing dietary patterns in Latin American societies are affecting what young children are fed. There is also a need to better understand potential disparities of infant feeding among indigenous and non-indigenous populations. In nearly all case study countries, key informants indicated that children from indigenous communities are generally worse off in terms of dietary quality and their households have difficulties in accessing a diverse diet. Local customs, beliefs and traditions also pose a key barrier for healthy diets.

Services in the dimensions of social protection, health care and WASH can and do contribute to better nutrition for children, but further efforts in these sectors are needed. For example, there is a wide gap in access to safe water and sanitation between urban and rural households and among different countries [32], with Bolivia and Haiti having particularly worrying conditions. According to key informants, households can easily access health services, although geographic distance remains a barrier, especially for those in rural and remote areas. Another barrier in the health system appears to be the lack of empathy or the vertical relation between health care personnel and parents. Due to these issues, parents may have less trust in the formal health system and turn to relatives or other acquaintances for advice and guidance on feeding practices. The role of health services will become more important as LAC embarks on health systems reforms to achieve universal health coverage. The region is currently characterized by large inequalities in access to quality health services [54], and health systems in the region need to strengthen their ties with other sectors to act on the social determinants of health and the risk factors associated with non-communicable diseases, including undernutrition and overweight [55]. The main bottleneck for social protection services is coverage. Most programmes are targeted to rural populations, while urbanization and urban poverty are increasing. For example, the coverage rates for all types of social assistance programmes among the poorest 20% rural households was 72%, 86% 93% and 87% in Guatemala, Paraguay, Peru and Uruguay respectively. For urban households in the bottom quintile, these rates were 63%, 85%, 80% and 90% in the respective countries. Coverage gaps are wider for the total population [35].

Large-scale national policies to reduce micronutrient deficiencies and underweight are implemented in the majority of LAC countries, with supplementation and staple food fortification programmes being the most popular. Despite national implementation, the effectiveness of these programmes depends on the coverage level, which in many countries leaves room for

improvement. In addition, several countries in the region could benefit from fortifying oil with vitamin A as a new strategy, including Argentina, Brazil, Dominican Republic, Ecuador, Guatemala, Honduras, Jamaica, Mexico, Paraguay and Suriname [56,57].

New efforts are emerging to reduce the triple burden of malnutrition, for instance in the form of sugar taxes, marketing restrictions and labeling requirements. However, inter-sectoral and comprehensive action against overweight and obesity remains a rarity, and more evidence is needed about the effectiveness of individual policies as well as cross-cutting strategies. This was echoed in the qualitative interviews, as key informants called for more integrated and cross-sectoral programming to address nutritional problems. A key theme was the issue of the level of coordination, as for example in Peru, major advancements were made when the authority for the nutrition agenda was at the prime minister's level, rather than at the level of a line ministry. Perhaps the most pressing gap is related to tackling overweight and obesity. Since these are emerging health challenges in the region, recent reviews found relevant strategic action to be weak or uncoordinated [37,39]. While the initiatives to reduce the desirability of unhealthy foods is a good start, more comprehensive, intersectoral action may be necessary to overcome the triple burden of malnutrition.

The knowledge, attitudes and practices of caregivers vary, and programmes aimed at improving infant feeding practices should consider the local preferences, beliefs, and socio-cultural contexts, as well as the interactions between service providers and caregivers. In the qualitative interviews, several informants noted the lack of awareness on adequate feeding practices among caregivers. They called for additional programming to counsel caregivers in complementary feeding practices, as well as improve the interaction between service providers and caregivers. A notable example is the development and implementation of the food guides in Paraguay, which are used to demonstrate the composition of a healthy diet for children under two years. Understanding the strengths, opportunities as well as the challenges of different population groups' infant feeding preferences may prove useful in designing infant and young child feeding promotion strategies that caregivers can comply with. Undertaking formative research in the community before implementation was found to be a cornerstone of culturally relevant nutrition education programmes in Peru [58]. Such an iterative process in programme design was also crucial to the strengthening and successful scale-up of the Integrated Strategy for Attention to Nutrition element of Mexico's flagship CCT programme [59].

This study has several limitations. The literature review was not a systematic review and hence, some relevant studies could have been overlooked. However, the included studies paint a comprehensive picture for the LAC region and for some countries in particular. The triangulation with key informants for selected countries further strengthened the completeness of the information. As for the qualitative part of the study, the number of key informants per country was limited, which may have led to relatively narrow views. The relatively small sample of four countries may prevent major generalizations from these findings, although care was taken to select countries with varying nutritional challenges and socio-economic conditions. In addition, informants were purposefully selected through local UNICEF offices, which may have also influenced the results. Nevertheless, the interviewers were independent researchers and key themes per country were triangulated between interviews to draw out recurrent insights.

Overall, the regional analysis showed that children's diets in LAC are shaped by several important drivers. Most notably, poverty and inequality, unequal access to services, inadequate coverage of social programmes and lack of awareness on appropriate feeding practices. In addition, disparities within countries by wealth and residence are important factors that influence children's diets. Countries in the region need to invest in policies to tackle all forms of malnutrition in young children, considering inequalities between and within countries, enhance coverage of social protection programmes, improve coordination between sectors to

improve children's diets and expand coverage and intensity of awareness campaigns on feeding practices, using iterative programme designs.

## Supporting information

**S1 Text. Literature review search strategy.**
(DOCX)

**S2 Text. Figures with disaggregated data.**
(DOCX)

**S3 Text. SRQR checklist.**
(DOCX)

**S4 Text. Questionnaire on inclusivity.**
(DOCX)

**S1 Data. Data Figs 2–4.**
(ZIP)

## Acknowledgments

We are grateful for comments from Jessica White (UNICEF New York) on an earlier version of the report. We also thank UNICEF colleagues in the case study countries for their valuable assistance and coordination: Maria Claudia Santizo from Guatemala, Sonia Avalos from Paraguay, Maria Elena Ugaz from Peru and Nora D'Oliveira from Uruguay. Finally, we are indebted to the key informants who generously gave their time to talk to us and share their insights.

## Author Contributions

**Conceptualization:** Franziska Gassmann, Richard de Groot, Stephan Dietrich, Eszter Timar, Yvette Fautsch-Macías, Paula Veliz, Aashima Garg, Maaike Arts.

**Data curation:** Eszter Timar, Florencia Jaccoud, Lorena Giuberti.

**Formal analysis:** Richard de Groot, Stephan Dietrich, Eszter Timar, Giulio Bordon.

**Methodology:** Franziska Gassmann, Eszter Timar.

**Supervision:** Franziska Gassmann, Yvette Fautsch-Macías, Paula Veliz, Maaike Arts.

**Writing – original draft:** Franziska Gassmann, Richard de Groot, Stephan Dietrich, Eszter Timar, Giulio Bordon.

**Writing – review & editing:** Franziska Gassmann, Richard de Groot, Yvette Fautsch-Macías, Paula Veliz, Aashima Garg, Maaike Arts.

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
