## [Decision Letter · Decision Letter 0]

11 Nov 2021

PGPH-D-21-00814

Determinants and drivers of young children’s diets in Latin America and the Caribbean: Findings from a regional analysis

Dear Dr. Richard de Groot,

Thank you very much for submitting your manuscript to PLOS Global Public Health. We have considered the paper carefully and feel that the paper addresses an important topic and has merit. However, the reviewers made valuable revision suggestions to help make it a stronger paper.Therefore, we invite you to submit a revised version of the manuscript that addresses the points raised by the reviewers (listed at the end of this email).

We look forward to receiving your revised manuscript.

Kind regards,

Bai Li, PhD

Academic Editor

2. Thank you for stating "Please note that ethical approval for this type of study was not required by our institutions." Please clarify whether your ethics committee specifically granted you a waiver of ethics approval.

3. Please provide additional details regarding participant consent for interviews. In the ethics statement in the Methods and online submission information, please ensure that you have specified:

 - whether consent was obtained

 - whether consent was informed 

 - what type of consent you obtained (for instance, written or verbal, and if verbal, how it was documented and witnessed). 

 - if the need for consent was waived by the ethics committee, please include this information.

4. Please provide separate figure files in .tif or .eps format only, and remove any figures embedded in your manuscript file.  If you are using LaTeX, you do not need to remove embedded figures.

5. In the online submission form, you indicated that "Data used in the manuscript to construct Figures 2-4 are publicly available through data.unicef.org. Primary data collection for this study involved in-depth interviews with selected key informants. Since transcripts of these interviews contain potentially identifying information, these transcripts will only be made available upon request."

6. Pleas eamend your detailed Financial Disclosure statement. This is published with the article, therefore should be completed in full sentences and contain the exact wording you wish to be published.

i). State the initials, alongside each funding source, of each author to receive each grant.

ii). State what role the funders took in the study. If the funders had no role in your study, please state: “The funders had no role in study design, data collection and analysis, decision to publish, or preparation of the manuscript.”

Reviewers' comments:

Reviewer's Responses to Questions

**Comments to the Author**

1. Does this manuscript meet PLOS Global Public Health’s publication criteria? Is the manuscript technically sound, and do the data support the conclusions? The manuscript must describe methodologically and ethically rigorous research with conclusions that are appropriately drawn based on the data presented.

Reviewer #1: Yes

Reviewer #2: Partly

2. Has the statistical analysis been performed appropriately and rigorously?

Reviewer #1: N/A

Reviewer #2: No

3. Have the authors made all data underlying the findings in their manuscript fully available (please refer to the Data Availability Statement at the start of the manuscript PDF file)?

Reviewer #1: Yes

Reviewer #2: Yes

4. Is the manuscript presented in an intelligible fashion and written in standard English?

Reviewer #1: Yes

Reviewer #2: No

5. Review Comments to the Author

Reviewer #1: In this article, the authors used a mixed approach to analyze the determinants and drivers of young children’s diets in Latin American and Caribbean (LAC) region. Both desk-based literature review and qualitative interviews with key informants in four LAC countries were conducted. Their findings well summarized the key determinants and drivers for inadequate diets in LAC regions and could be used as cues for program/policy design to tackle the malnutritional problem. However, there are some major and minor concerns that should be addressed before the manuscript can be published.

First, the reason why the four countries (Guatemala, Paraguay, Peru, and Uruguay) were chosen for the key informant interviews needs to be stated in a clearer way. Do they sufficiently represent the dietary situation of young children in the LAC region?

Second, according to the authors, analysis of key informants interview focused on extracting key emerging themes across six broad topics was performed (Line 133-138). However, in the result (Section 3.2.) only a summary of key informant interview per country was shown. The interview results of each country should be listed in a table and categorized based on each relevant topics, so that readers could compare the results across these countries and identify the common or unique drivers and determinants.

Third, in line 44, the authors defined that “undernutrition” includes “stunting, wasting, and micronutrient deficiencies”. However, in line 341 and 343, “undernutrition” and “micronutrient deficiencies” are listed on the same hierarchy. The definition of “undernutrition” should be clearly defined and kept consistent throughout the manuscript.

In addition, there are some minor concerns:

1) Figure legends are missing. It is difficult to understand the abbreviations in Fig 2-4.

2) The color contrast for Figure 2 and 4 is not strong. Especially in figure 4, it’s difficult to determine whether the bar charts of “BOL 2016” and “PER 2016” represent “minimum diet diversity” or “minimum acceptable diet”. The color for “minimum diet diversity” or “minimum acceptable diet” is too close to be distinguished.

3) Line 28-29, the citation of this statement should be included.

4) Line 62, the full name of “ANC” should be showed when this abbreviation appears at the first time.

Reviewer #2: ------------------------------------------------------

Summary and overall impression of the research

This article aimed to identify determinants and drivers of children’s diets in the Latin American and Caribbean (LAC) region with a regional analysis. The authors address an important topic, especially given the current triple burden of malnutrition, current disparities, and budgetary constraints within the LAC region which urge for context-specific understanding and actions. This article holds much promise, and the authors are off to a good start as they employ multiple methods, and make use of a helpful framework to guide the identification of drivers, and structuring of recommended actions. Furthermore, they selected and interviewed key informants of different organisations and levels of influence, of multiple countries which differ in terms of socio-economic and demographic development within the LAC region. The authors share several very interesting and important contextual insights on drivers of children's diet, and actions for addressing the double (or triple) burden of malnutrition in the LAC region.

There are however multiple issues throughout this article which require major revision.

In the introduction, the authors make a strong argument for focussing on disparities within the LAC region, however there is limited and inconsistent continuation of this perspective in the remainder of the article. Most importantly, the formulated research aim does not fully capture all study components. The authors are advised to formulate a list of multiple sub-questions/objectives which represent all utilised study methods. Furthermore, some of the concepts such as determinants/drivers and children’s diet are not well-defined, and used interchangeably which makes it difficult for the reader to extrapolate the key findings and conclusions.

The methods section is clearly structured, and the analysis approach of key informant data is well-described. However, this section currently does not contain sufficient detail for the author’s study to be accurately replicated by other researchers. Furthermore, this article currently reads as a “multiple methods” study instead of a “mixed-methods” study. There is currently no integration, merging or embedding of the different data sources. Also, a more detailed description on data and methodological triangulation processes would increase the credibility and validity of research findings. Lastly, the process of obtaining ethical consent currently lacks detail.

The overall lack detail in the methods section translates into a fairly chaotic presentation of the results and discussion on determinants and drivers of children’s diets. It is unclear which type of literature they focussed on and whether the results of the literature review are exhaustive. Tables which summarise the selected literature would be helpful here to provide context about the presentation of findings, as it is not clear what studies were identified and in relation to which outcomes. Furthermore, the authors should consider only presenting their own findings in the results section. They’re currently referring to other literature in the results section which is preferably positioned in the introduction/discussion section.

While the summarised conclusions are succinct, these only address part of the aim and findings, and only refer to common themes. The conclusions on drivers of young children's diets, and implications for future actions also do not represent the disparities between and within countries in the LAC region.

Lastly, while this article was generally written in an intelligible fashion and in standard English, it could still benefit from proof-reading before submitting the subsequent version as ambiguous language was used in multiple sections.

Please find below a more detailed list of the major and minor issues for the authors to consider and address in the subsequent version:

--------------

Major issues

--------------

1. Abstract

1.1. The authors present the scope of the problem, however it would be helpful if they also introduce child feeding which is currently not well-represented in the abstract.

1.2. Please state in the abstract that you’ve conducted a descriptive analysis.

1.3. The results and conclusions focus on key themes, however not all of these are common, based on the findings. There is limited information on disparities between and within countries for nutrition, and diet outcomes and related determinants/drivers.

1.4. The concepts presented in the key findings would benefit from more detail. For example, from reading the abstract it is unclear what type of “inequality” the authors refer to.

2. Introduction

2.1. The introduction is missing a section which signifies the need for this study. While the authors do a good job of setting the stage, a section which summarises the latest evidence, and how their work is filling a gap would be helpful here. There is also no rationale for the need of a literature review, or this is hidden in your wording. Please make this explicit in your introduction.

2.2. It is currently unclear what the authors mean with “determinants” and “drivers”. Please clarify the difference in the introduction. This terminology is also not consistently used in the remainder of the article.

2.3. The research aim is not comprehensive in its current state. For example, the aim does not capture the author’s descriptive analysis. Consider formulating a list of multiple sub-questions/objectives which link to the used methods.

2.4. Line 38, the authors state they used a mixed-methods approach but do not explicitly state what they mean by this. Based on the analysis and presentation of the findings this is a multiple methods study. There is no merging, integrating or embedding of the different research methods.

3. Methods

Descriptive analysis

3.1. It is currently unclear where the authors retrieved the malnutrition rates from. The reference directs me to child feeding statistics. Please clarify this.

3.2. The methods section currently misses a paragraph which provides a justification for the selected data sources for the descriptive analysis. It would be helpful to present an overview of current publicly available data sources on child nutrition and diets.

3.3. Please specify which software and commands you’ve used to conduct the descriptive analysis.

Literature review

3.4. It is currently unclear what type of desk-based literature review is conducted. Please clarify this, and the appropriate supplementary materials including a guideline (e.g. PRISMA).

3.5. Please list all visited databases and used search terms so that others will be able to replicate this review. It would also be helpful to supplement a table which includes the applied search strategy. A table which includes the used term translations would also be very helpful for future research.

3.6. Please clarify which types of academic literature you included (e.g., quantitative, qualitative, mixed-methods)

Key informant interviews

3.7. Please provide more detail on the interviewing process. Will there be any guides attached for the reader to access? It would be valuable to see how the guides differed based on the stakeholder and supplement these with the article.

3.8. The description of the key informant interviews requires more detail. Currently, a section is missing on the recruitment process of key informants, and the taken quality measures. What did the authors do to enhance the trustworthiness of the findings (e.g., member checking, triangulation, reflexivity). Furthermore who translated the transcripts, and were any cross-translations conducted?

3.9. The authors mentioned that the analysis focussed on extracting key emerging themes across six broad topics. It would be helpful for the reader when they specify the analysis technique and include the relevant literature (e.g., thematic analysis with a framework?).

3.10. The reporting of key informant interviews should be accompanied with a Consolidated criteria for reporting qualitative research (COREQ) checklist or Standards for reporting qualitative research (SRQR) checklist.

4. Results

Descriptive statistics

4.1. I advise the authors to edit the figures by including footnotes which spell out the country names, and insert the appropriate legend labels. The statistics provided in the figures could benefit from stratification by different residential types, socio-economic status of households, and other demographic characteristics if possible.

Literature review

4.2. While this the literature review section focuses on drivers/determinants of children’s diet, the authors often refer to children’s (sometimes not mentioning this either) nutrition status which is not proposed as a study outcome in the research aim. The focus of this section should align with the proposed research aim which is to identified determinants of children’s diets as an outcome of determinants/drivers according to the UNICEF framework.

4.3. Please present the type of methods of your selected studies in appropriate tables so that the reader can assess the strength of evidence on the determinants of children’s diets. It would also be helpful if the selected tables are stratified by country and country groups based on socio-economic development (criteria used for the selection of countries for the key informant interviews)

4.4. It would greatly strengthen your analysis when you conduct a cross-comparison analysis on the different countries to bring out the disparities on drivers and actions.

4.5. Information on prevalence rates, and other statistics should either be positioned in the introduction (or discussion) to provide context, unless the identified studies presented evidence on increasing rates (for example infant/child milk-based formulas) as key determinants of children’s diets.

4.6. The authors often make general statements about countries in the region (for example, line 235). It would be very helpful when the authors specify the countries.

Key informant interviews

4.7. Contextual information for each country should ideally be positioned in the methods section (statistics on child undernutrition) when this is not a direct result of your key informant interview.

4.8. In several sections it is unclear whether a statement represents the view the a key informant, or whether this a general statement derived from literature or made by the author. Please rephrase so that this is clearer.

4.9. Consider including participant's quotes, and highlighting different and contrasting views of the key informants of each country.

4.10. The results and discussion sections which cover the key informant’s perspectives could benefit from more detail if the data allows this. For example, it would be valuable to know why there is currently a lack of inter-institutional collaboration and continuity over time in Government’s actions in [country] (line 308) , and more detail on why more progress was made under the prime minister’s responsibility (line 320) to give a few examples. This could really enrich the findings and discussion.

5. Discussion

5.1. The discussion does not contain a section on the strengths and limitations of this study. Please include this in the subsequent version.

5.2. Line 376 – are these new results? I could not locate these statements in the results. Please double-check the discussion to ensure that no new findings are introduced.

5.3. The discussion in some sections is diverging from child nutrition, it is unclear at times what the outcome of interest is. Children’s diets? Or child nutrition status? Please address this. Furthermore, much of the selected literature is not specified to children within the set age-range. This makes it difficult to allocate any of the identified drivers/determinants specific to that age group.

5.4. Some statements related to the key informant views are not supported by the findings. For example, the authors state that healthy food is more difficult to access, and more expensive, this is however not reported for all countries, and phrased differently. It would be helpful if the authors double-check this in their subsequent version. Also, please specify which countries you’re referring to when making these statements. Was this for example also mentioned for Peru? It would strengthen your findings and discussion when you highlight the differences between these countries and interpret what this means for future actions in the region.

5.5. In the discussion some countries are mentioned as examples, but it is not clear what and how the disparities are coming forward. It would greatly strengthen this article when the authors discuss these in more detail and interpret what this means for context-specific drivers and actions.

--------------

Minor issues

--------------

1. Abstract

1.1. It is unclear in some parts which age group the author’s focus on. It would be helpful for the reader when they’re informed about this when reading the abstract.

2. Introduction

2.1. Please provide more detail on the introduced concepts. For example, it would be helpful to provide definitions of the malnutrition indicators and children’s diets in the introduction.

2.2. Consider providing statistics on all forms of child nutrition, including overweight and obesity. It is well documented that most regions are facing double/triple burdens of malnutrition so my advice is to already introduce this phenomenon in the introduction and refer to the latest Lancet series on double burden of malnutrition for example.

3. Methods

3.1. Line 89 – What is meant by “secondary data analysis”? Please clarify this.

3.2. Please clarify and describe the used snowballing techniques for identif.

3.3. Did the authors use any BOOLEAN terms for your literature search? Consider stating this.

4. Results

4.1. The descriptive analysis results miss statistics on obesity. If these are not identifiable it is safer to just state overweight within-text.

4.2. The identified literature which relates to access of food, appropriate services and adequate practices is currently presented across different levels of influence (e.g., individual, household and macro-environmental levels). Consider making this explicit in your writing to make it easier for the readers to follow the narrative.

4.3. Please provide a definition of “access to adequate food”, does this also include economic and physical dimensions?

4.4. Line 173 – Consider avoiding ambiguous words such as “encouraging” and “striking” as this is open to multiple interpretations.

4.5. Line 205 – this is a very interesting comparison. Consider however positioning this in the discussion. Since you’re looking at disparities within the LAC region, you could you also tailor this example to the price of a similar plate in a higher-middle income country in the LAC region which then highlights the disparities within the region.

4.6. Line 215 – Is there any literature available on socio-economic differences within rural and urban areas for animal source food consumption? Please also refer to the literature.

4.7. Some sections in the results would benefit from more fluent transitions. For example, the identified qualitative research (refreshing that you include this type of literature!) on WASH services does not build on the previous paragrap and feels therefore isolated (line 238). Please connect this to the narrative.

4.8. Line 256 – Which policies are being referred to here? Please specify this.

4.9. Please position the table which includes the characteristics of the key informants in the Results section.

4.10. Line 284 – The rise of sugar consumption is probably not the sole reason for adoption of regulation strategies, please consider rephrasing this.

5. Discussion

5.1. Line 333 – What is meant by a “landscape analysis”? I advise not introducing new terminology in the discussion. Or introduce this type of analysis in the methods section alongside the appropriate literature.

5.2. Lines 338-344 – it is unclear whether this section refers to the descriptive analysis. Please clarify this.

5.3. New concepts and development trends are introduced, such as the triple burden of malnutrition (line 343) and dietary transition (line 338), consider presenting these in the introduction. These are well-established and backed up by a rich literature database.

5.4. Line 421 – Adopting an iterative process to programme design could be a key conclusion, however the results do not cover this strongly.

6. PLOS authors have the option to publish the peer review history of their article (what does this mean?). If published, this will include your full peer review and any attached files.

**Do you want your identity to be public for this peer review?** For information about this choice, including consent withdrawal, please see our Privacy Policy.

Reviewer #1: No

Reviewer #2: No

---

## [Editor Report · Decision Letter 1]

4 Apr 2022

PGPH-D-21-00814R1

Determinants and drivers of young children’s diets in Latin America and the Caribbean: Findings from a regional analysis

Dear Dr. de Groot,

Thank you for submitting your manuscript to PLOS Global Public Health. After careful consideration, we feel that it has merit but does not fully meet PLOS Global Public Health’s publication criteria as it currently stands. Therefore, we invite you to submit a revised version of the manuscript that addresses the points raised during the review process (please see the end of this email).

We look forward to receiving your revised manuscript.

Kind regards,

Bai Li, PhD

Academic Editor

**Reviewers' comments:**

**Reviewer 1:**

In this article, the authors used a mixed approach to analyze the determinants and drivers of young children’s diets in Latin American and Caribbean (LAC) region. Both desk-based literature review and qualitative interviews with key informants in four LAC countries were conducted. Their findings well summarized the key determinants and drivers for inadequate diets in LAC regions and could be used as cues for program/policy design to tackle the malnutritional problem. However, there are some major and minor concerns that should be addressed before the manuscript can be published.

First, the reason why the four countries (Guatemala, Paraguay, Peru, and Uruguay) were chosen for the key informant interviews needs to be stated in a clearer way. Do they sufficiently represent the dietary situation of young children in the LAC region?

Second, according to the authors, analysis of key informants interview focused on extracting key emerging themes across six broad topics was performed (Line 133-138). However, in the result (Section 3.2.) only a summary of key informant interview per country was shown. The interview results of each country should be listed in a table and categorized based on each relevant topics, so that readers could compare the results across these countries and identify the common or unique drivers and determinants.

Third, in line 44, the authors defined that “undernutrition” includes “stunting, wasting, and micronutrient deficiencies”. However, in line 341 and 343, “undernutrition” and “micronutrient deficiencies” are listed on the same hierarchy. The definition of “undernutrition” should be clearly defined and kept consistent throughout the manuscript.

In addition, there are some minor concerns:

1) Figure legends are missing. It is difficult to understand the abbreviations in Fig 2-4.

2) The color contrast for Figure 2 and 4 is not strong. Especially in figure 4, it’s difficult to determine whether the bar charts of “BOL 2016” and “PER 2016” represent “minimum diet diversity” or “minimum acceptable diet”. The color for “minimum diet diversity” or “minimum acceptable diet” is too close to be distinguished.

3) Line 28-29, the citation of this statement should be included.

4) Line 62, the full name of “ANC” should be showed when this abbreviation appears at the first time.

**Reviewer 2:**

Summary and overall impression of the research

This article aimed to identify determinants and drivers of children’s diets in the Latin American and Caribbean (LAC) region with a regional analysis. The authors address an important topic, especially given the current triple burden of malnutrition, current disparities, and budgetary constraints within the LAC region which urge for context-specific understanding and actions. This article holds much promise, and the authors are off to a good start as they employ multiple methods, and make use of a helpful framework to guide the identification of drivers, and structuring of recommended actions. Furthermore, they selected and interviewed key informants of different organisations and levels of influence, of multiple countries which differ in terms of socio-economic and demographic development within the LAC region. The authors share several very interesting and important contextual insights on drivers of children's diet, and actions for addressing the double (or triple) burden of malnutrition in the LAC region.

There are however multiple issues throughout this article which require major revision.

In the introduction, the authors make a strong argument for focussing on disparities within the LAC region, however there is limited and inconsistent continuation of this perspective in the remainder of the article. Most importantly, the formulated research aim does not fully capture all study components. The authors are advised to formulate a list of multiple sub-questions/objectives which represent all utilised study methods. Furthermore, some of the concepts such as determinants/drivers and children’s diet are not well-defined, and used interchangeably which makes it difficult for the reader to extrapolate the key findings and conclusions.

The methods section is clearly structured, and the analysis approach of key informant data is well-described. However, this section currently does not contain sufficient detail for the author’s study to be accurately replicated by other researchers. Furthermore, this article currently reads as a “multiple methods” study instead of a “mixed-methods” study. There is currently no integration, merging or embedding of the different data sources. Also, a more detailed description on data and methodological triangulation processes would increase the credibility and validity of research findings. Lastly, the process of obtaining ethical consent currently lacks detail.

The overall lack detail in the methods section translates into a fairly chaotic presentation of the results and discussion on determinants and drivers of children’s diets. It is unclear which type of literature they focussed on and whether the results of the literature review are exhaustive. Tables which summarise the selected literature would be helpful here to provide context about the presentation of findings, as it is not clear what studies were identified and in relation to which outcomes. Furthermore, the authors should consider only presenting their own findings in the results section. They’re currently referring to other literature in the results section which is preferably positioned in the introduction/discussion section.

While the summarised conclusions are succinct, these only address part of the aim and findings, and only refer to common themes. The conclusions on drivers of young children's diets, and implications for future actions also do not represent the disparities between and within countries in the LAC region.

Lastly, while this article was generally written in an intelligible fashion and in standard English, it could still benefit from proof-reading before submitting the subsequent version as ambiguous language was used in multiple sections.

Please find below a more detailed list of the major and minor issues for the authors to consider and address in the subsequent version:

--------------

Major issues

--------------

1. Abstract

1.1. The authors present the scope of the problem, however it would be helpful if they also introduce child feeding which is currently not well-represented in the abstract.

1.2. Please state in the abstract that you’ve conducted a descriptive analysis.

1.3. The results and conclusions focus on key themes, however not all of these are common, based on the findings. There is limited information on disparities between and within countries for nutrition, and diet outcomes and related determinants/drivers.

1.4. The concepts presented in the key findings would benefit from more detail. For example, from reading the abstract it is unclear what type of “inequality” the authors refer to.

2. Introduction

2.1. The introduction is missing a section which signifies the need for this study. While the authors do a good job of setting the stage, a section which summarises the latest evidence, and how their work is filling a gap would be helpful here. There is also no rationale for the need of a literature review, or this is hidden in your wording. Please make this explicit in your introduction.

2.2. It is currently unclear what the authors mean with “determinants” and “drivers”. Please clarify the difference in the introduction. This terminology is also not consistently used in the remainder of the article.

2.3. The research aim is not comprehensive in its current state. For example, the aim does not capture the author’s descriptive analysis. Consider formulating a list of multiple sub-questions/objectives which link to the used methods.

2.4. Line 38, the authors state they used a mixed-methods approach but do not explicitly state what they mean by this. Based on the analysis and presentation of the findings this is a multiple methods study. There is no merging, integrating or embedding of the different research methods.

3. Methods

Descriptive analysis

3.1. It is currently unclear where the authors retrieved the malnutrition rates from. The reference directs me to child feeding statistics. Please clarify this.

3.2. The methods section currently misses a paragraph which provides a justification for the selected data sources for the descriptive analysis. It would be helpful to present an overview of current publicly available data sources on child nutrition and diets.

3.3. Please specify which software and commands you’ve used to conduct the descriptive analysis.

Literature review

3.4. It is currently unclear what type of desk-based literature review is conducted. Please clarify this, and the appropriate supplementary materials including a guideline (e.g. PRISMA).

3.5. Please list all visited databases and used search terms so that others will be able to replicate this review. It would also be helpful to supplement a table which includes the applied search strategy. A table which includes the used term translations would also be very helpful for future research.

3.6. Please clarify which types of academic literature you included (e.g., quantitative, qualitative, mixed-methods)

Key informant interviews

3.7. Please provide more detail on the interviewing process. Will there be any guides attached for the reader to access? It would be valuable to see how the guides differed based on the stakeholder and supplement these with the article.

3.8. The description of the key informant interviews requires more detail. Currently, a section is missing on the recruitment process of key informants, and the taken quality measures. What did the authors do to enhance the trustworthiness of the findings (e.g., member checking, triangulation, reflexivity). Furthermore who translated the transcripts, and were any cross-translations conducted?

3.9. The authors mentioned that the analysis focussed on extracting key emerging themes across six broad topics. It would be helpful for the reader when they specify the analysis technique and include the relevant literature (e.g., thematic analysis with a framework?).

3.10. The reporting of key informant interviews should be accompanied with a Consolidated criteria for reporting qualitative research (COREQ) checklist or Standards for reporting qualitative research (SRQR) checklist.

4. Results

Descriptive statistics

4.1. I advise the authors to edit the figures by including footnotes which spell out the country names, and insert the appropriate legend labels. The statistics provided in the figures could benefit from stratification by different residential types, socio-economic status of households, and other demographic characteristics if possible.

Literature review

4.2. While this the literature review section focuses on drivers/determinants of children’s diet, the authors often refer to children’s (sometimes not mentioning this either) nutrition status which is not proposed as a study outcome in the research aim. The focus of this section should align with the proposed research aim which is to identified determinants of children’s diets as an outcome of determinants/drivers according to the UNICEF framework.

4.3. Please present the type of methods of your selected studies in appropriate tables so that the reader can assess the strength of evidence on the determinants of children’s diets. It would also be helpful if the selected tables are stratified by country and country groups based on socio-economic development (criteria used for the selection of countries for the key informant interviews)

4.4. It would greatly strengthen your analysis when you conduct a cross-comparison analysis on the different countries to bring out the disparities on drivers and actions.

4.5. Information on prevalence rates, and other statistics should either be positioned in the introduction (or discussion) to provide context, unless the identified studies presented evidence on increasing rates (for example infant/child milk-based formulas) as key determinants of children’s diets.

4.6. The authors often make general statements about countries in the region (for example, line 235). It would be very helpful when the authors specify the countries.

Key informant interviews

4.7. Contextual information for each country should ideally be positioned in the methods section (statistics on child undernutrition) when this is not a direct result of your key informant interview.

4.8. In several sections it is unclear whether a statement represents the view the a key informant, or whether this a general statement derived from literature or made by the author. Please rephrase so that this is clearer.

4.9. Consider including participant's quotes, and highlighting different and contrasting views of the key informants of each country.

4.10. The results and discussion sections which cover the key informant’s perspectives could benefit from more detail if the data allows this. For example, it would be valuable to know why there is currently a lack of inter-institutional collaboration and continuity over time in Government’s actions in [country] (line 308) , and more detail on why more progress was made under the prime minister’s responsibility (line 320) to give a few examples. This could really enrich the findings and discussion.

5. Discussion

5.1. The discussion does not contain a section on the strengths and limitations of this study. Please include this in the subsequent version.

5.2. Line 376 – are these new results? I could not locate these statements in the results. Please double-check the discussion to ensure that no new findings are introduced.

5.3. The discussion in some sections is diverging from child nutrition, it is unclear at times what the outcome of interest is. Children’s diets? Or child nutrition status? Please address this. Furthermore, much of the selected literature is not specified to children within the set age-range. This makes it difficult to allocate any of the identified drivers/determinants specific to that age group.

5.4. Some statements related to the key informant views are not supported by the findings. For example, the authors state that healthy food is more difficult to access, and more expensive, this is however not reported for all countries, and phrased differently. It would be helpful if the authors double-check this in their subsequent version. Also, please specify which countries you’re referring to when making these statements. Was this for example also mentioned for Peru? It would strengthen your findings and discussion when you highlight the differences between these countries and interpret what this means for future actions in the region.

5.5. In the discussion some countries are mentioned as examples, but it is not clear what and how the disparities are coming forward. It would greatly strengthen this article when the authors discuss these in more detail and interpret what this means for context-specific drivers and actions.

--------------

Minor issues

--------------

1. Abstract

1.1. It is unclear in some parts which age group the author’s focus on. It would be helpful for the reader when they’re informed about this when reading the abstract.

2. Introduction

2.1. Please provide more detail on the introduced concepts. For example, it would be helpful to provide definitions of the malnutrition indicators and children’s diets in the introduction.

2.2. Consider providing statistics on all forms of child nutrition, including overweight and obesity. It is well documented that most regions are facing double/triple burdens of malnutrition so my advice is to already introduce this phenomenon in the introduction and refer to the latest Lancet series on double burden of malnutrition for example.

3. Methods

3.1. Line 89 – What is meant by “secondary data analysis”? Please clarify this.

3.2. Please clarify and describe the used snowballing techniques for identif.

3.3. Did the authors use any BOOLEAN terms for your literature search? Consider stating this.

4. Results

4.1. The descriptive analysis results miss statistics on obesity. If these are not identifiable it is safer to just state overweight within-text.

4.2. The identified literature which relates to access of food, appropriate services and adequate practices is currently presented across different levels of influence (e.g., individual, household and macro-environmental levels). Consider making this explicit in your writing to make it easier for the readers to follow the narrative.

4.3. Please provide a definition of “access to adequate food”, does this also include economic and physical dimensions?

4.4. Line 173 – Consider avoiding ambiguous words such as “encouraging” and “striking” as this is open to multiple interpretations.

4.5. Line 205 – this is a very interesting comparison. Consider however positioning this in the discussion. Since you’re looking at disparities within the LAC region, you could you also tailor this example to the price of a similar plate in a higher-middle income country in the LAC region which then highlights the disparities within the region.

4.6. Line 215 – Is there any literature available on socio-economic differences within rural and urban areas for animal source food consumption? Please also refer to the literature.

4.7. Some sections in the results would benefit from more fluent transitions. For example, the identified qualitative research (refreshing that you include this type of literature!) on WASH services does not build on the previous paragrap and feels therefore isolated (line 238). Please connect this to the narrative.

4.8. Line 256 – Which policies are being referred to here? Please specify this.

4.9. Please position the table which includes the characteristics of the key informants in the Results section.

4.10. Line 284 – The rise of sugar consumption is probably not the sole reason for adoption of regulation strategies, please consider rephrasing this.

5. Discussion

5.1. Line 333 – What is meant by a “landscape analysis”? I advise not introducing new terminology in the discussion. Or introduce this type of analysis in the methods section alongside the appropriate literature.

5.2. Lines 338-344 – it is unclear whether this section refers to the descriptive analysis. Please clarify this.

5.3. New concepts and development trends are introduced, such as the triple burden of malnutrition (line 343) and dietary transition (line 338), consider presenting these in the introduction. These are well-established and backed up by a rich literature database.

5.4. Line 421 – Adopting an iterative process to programme design could be a key conclusion, however the results do not cover this strongly.

---

## [Editor Report · Decision Letter 2]

9 Jun 2022

Determinants and drivers of young children’s diets in Latin America and the Caribbean: Findings from a regional analysis

PGPH-D-21-00814R2

Dear Dr. de Groot,

We are pleased to inform you that your manuscript 'Determinants and drivers of young children’s diets in Latin America and the Caribbean: Findings from a regional analysis' has been provisionally accepted for publication in PLOS Global Public Health.

Best regards,

Bai Li, PhD

Academic Editor
